# State Feedback Stabilization for a Class of Upper-Triangular Stochastic Nonlinear Systems with Time-Varying Control Coefficients

**Xixi Sun, Haisheng Yu * and Xiaoyu Xu**

School of Mathematical and Statistical Sciences, Ludong University, Yantai 264025, China; sun19980203@163.com (X.S.); xxy18454401536@163.com (X.X.)

* Correspondence: haisenltn@163.com

**Abstract:** The problem explored in this article concerns the stability of the state feedback control of the upper-triangular stochastic nonlinear systems whose control coefficients are time-varying. First, the state feedback control of the corresponding nominal system is carried out by utilizing the backstepping technique combined with the appropriate Lyapunov function. Then, low-gain homogeneous domination technology and the efficient coordinate transformation method are adopted to realize the state feedback control of the original system and ensure the global asymptotic stability (GAS) in probability of the system. Finally, an example is given to illustrate the feasibility and correctness of the method.

**Keywords:** upper-triangular stochastic systems; state feedback; homogeneous domination

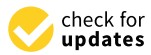



## 1. Introduction

The stability of stochastic systems occupies an essential place in the field of nonlinear control. On the one hand, it has an important theoretical value [1–9]. On the other hand, it has a strong practical significance [10–12]. It is worth noting that many physical systems such as dynamic ship positioning systems [10], robotic manipulators on seaborne platforms [11], and wireless sensor networks [12] can all be described by stochastic systems. In the past few years, many research works have been published on stochastic nonlinear systems [13–23], in which a quadratic or a quartic Lyapunov function is usually designed to tackle the stability problem of stochastic nonlinear systems.

Stochastic systems are commonly divided into two types, namely upper-triangular (i.e., feedforward) stochastic systems and lower-triangular (i.e., feedback) stochastic systems. In recent years, the control issues of lower-triangular stochastic nonlinear systems have become the focus of prominent research works with the assistance of the stochastic stability theory [24–31]. Among them, Refs. [29–31] are able to solve the stability problem of lower-triangular stochastic nonlinear systems through a homogeneous domination technology. The upper-triangle stochastic nonlinear systems have also attracted extensive attention due to its wide application in machinery and the aerospace industry. For example, the cart–pendulum system [32] and the vertical take-off and landing aircraft [33] can all be modeled as the upper-triangular structure. However, due to the particularity and complexity of the structure of the upper-triangular stochastic systems, many traditional methods such as the backstepping method are no longer applicable. This is because in the differential transformation of the Lyapunov function, stochastic differentiation produces a Hessian cross term for the diffusion term of the system, which makes the stability of the upper-triangular stochastic systems very challenging.

For upper-triangular systems, Ref. [34] introduces a scale gain into the controller to control the disturbance by using the homogeneous domination technology and generalizes this result to the upper-triangular stochastic nonlinear systems. Ref. [35] proposes a method

for tackling the stability problem on the basis of the homogeneous domination technique, i.e., the low-gain homogeneous domination technique. Ref. [36] then studies the stability problem of an upper-triangular stochastic system by utilizing the method for the first time. Furthermore, the stability problems of large-scale [37] and high-order [38] upper-triangular stochastic nonlinear systems as well as those disturbed by a second-order moment [39] are investigated by utilizing the low-gain homogeneous domination technique. However, we can notice that the systems in [36–39] are all with deterministic control coefficients. Compared with deterministic control coefficients, time-varying control coefficients have also received extensive attention [40], where only stochastic linear systems have been studied. The stability problem of upper-triangular stochastic nonlinear systems with time-varying control coefficients has not been researched.

On the basis of the above-mentioned literature, this paper utilizes the Lyapunov function and low-gain homogeneous domination technique to analyze the stability of the state feedback of the upper-triangular stochastic systems, which have time-varying control coefficients. This study includes at least two important contributions:

(i) The structure of the upper-triangular systems is complex. Therefore, in this article, a stability analysis of the nominal system is firstly carried out, and then the state feedback stability problem of the studied systems is solved by utilizing the low-gain homogeneous domination technique.

(ii) This paper considers general upper-triangular stochastic systems. This is in contrast to [36–39], which consider that the control coefficients of the systems are deterministic, while we study the systems with time-varying control coefficients. Time-varying control coefficients have uncertain upper and lower bounds, which, together with Young's inequality, ensure that the stability problem is solved.

The remaining parts are as follows. Section 2 presents relevant definitions. Section 3 discusses the main results. A numerical simulation is presented in Section 4. Finally, Section 5 brings the paper to a conclusion.

## 2. Relevant Definitions

The following symbols are commonly used in this paper. $R_+$ means the set of all nonnegative real numbers, and $R^n$ means the real $n$-dimensional space. $X^T$ is its transpose, $\text{Tr}\{X\}$ is recorded as its trace when $X$ is square, and $|X|$ indicates the 2-norm of vector $X$ in Euclidean space. For $A = A_{ij} \in R^{n \times m}$, defining $|A| = (\sum_{i=1}^{n} \sum_{j=1}^{m} A_{ij}^2)^{1/2}$ and $|A|_\infty = \max_{1 \le i \le n}\{\sum_{j=1}^{m} |A_{ij}|\}$. $C^i$ is recorded as the set of all functions with a continuous $i$th partial derivative. Class $\mathcal{K}$ indicates all of the $R_+ \to R_+$ functions that are continuous, strictly monotonic, and equal to zero at zero; class $\mathcal{K}_\infty$ indicates all the unbounded functions in $\mathcal{K}$; the function $\beta(s, t) \in \mathcal{KL} \colon R_+ \times R_+ \to R_+$ indicates that for a given $t$, $\beta(s, t) \in \mathcal{K}$, while for a given $s$, $\beta(s, t)$ is monotonically decreasing, and $\lim_{t \to \infty} \beta(s, t) = 0$.

For the stochastic nonlinear system

$$dx = f(x)dt + g^T(x)d\omega, \forall\, x_0 \in R^n, \tag{1}$$

where the state is $x \in R^n$, $\omega \in R^r$ is a Wiener process for independent standards defined in a probability space $(\Omega, \mathcal{F}, P)$. For any $t \ge 0$, when $x \in R^n$, the functions $f : R^n \to R^n$, $g^T : R^n \to R^{n \times r}$ are locally Lipschitz , and $f(0) = 0$, $g(0) = 0$. The following definitions are available.

**Definition 1** ([22]). *$\forall \varepsilon > 0$, $\exists \beta(\cdot, \cdot) \in \mathcal{KL}$, satisfying $P\{|x(t)| < \beta(|x_0|, t)\} \ge 1 - \varepsilon$, for $\forall t \ge 0$, $x_0 \in R^n \setminus \{0\}$, then system (1) is said to be globally asymptotically stable in probability, with an equilibrium point of $x = 0$.*

**Definition 2** ([35]). *For the coordinates $(x_1, \cdots, x_n) \in R^n$ and $0 < r_i \in R$, $i = 1, \cdots, n$:*
*(1) $\forall \varepsilon > 0$, defining the dilation $\triangle_\varepsilon(x) = (\varepsilon^{r_1} x_1, \cdots, \varepsilon^{r_n} x_n)$, where $r_i$ is the weight of the corresponding coordinates. For convenience, we define $\triangle = (r_1, \cdots, r_n)$;*

*(2) If $\tau \in R$, for $\forall x \in R^n \backslash \{0\}$, $\varepsilon > 0$, $V(\triangle_\varepsilon(x)) = \varepsilon^\tau V(x_1, \cdots, x_n)$, then the scalar function $V \in C(R^n, R)$ and the vector function $f \in C(R^n, R)$ are homogeneous of degree $\tau$;*

*(3) For $\forall x \in R^n$, such that $\|x\|_\triangle = (\sum_{i=1}^n |x_i|^{2/r_i})^{1/2}$.*

## 3. Main Results

Consider the following upper-triangular stochastic nonlinear systems in this paper:

$$
\begin{aligned}
dx_1 &= (d_1(t)x_2 + f_1(\widetilde{x}_3))dt + g_1^T(\widetilde{x}_3)d\omega, \\
dx_2 &= (d_2(t)x_3 + f_2(\widetilde{x}_4))dt + g_2^T(\widetilde{x}_4)d\omega, \\
&\vdots \\
dx_{n-2} &= (d_{n-2}(t)x_{n-1} + f_{n-2}(\widetilde{x}_n))dt + g_{n-2}^T(\widetilde{x}_n)d\omega, \\
dx_{n-1} &= d_{n-1}(t)x_n dt, \\
dx_n &= d_n(t)u dt,
\end{aligned}
\tag{2}
$$

where $x = (x_1, \cdots, x_n)^T \in R^n$ and $u \in R$ are respectively the system state and the input, while $\widetilde{x}_i = (x_i, \cdots, x_n)^T \in R^n$, $\omega \in R^r$ is a Wiener process for independent standards defined in a probability space $(\Omega, \mathcal{F}, P)$. For $i = 1, \cdots, n-2$, the functions $f_i : R^{n-i-1} \to R$ and $g_i : R^{n-i-1} \to R^r$ both vanish at the origin, and for $i = 1, \cdots, n$, $d_i(t) : R_+ \to R$ are unknown time-varying control coefficients with a known sign.

To study system (2), we make the following assumptions:

**Assumption 1.** *There is a constant $b > 0$ for $i = 1, \cdots, n$, such that the following equation holds:*

$$
\begin{aligned}
|f_i(\widetilde{x}_{i+2})| &\le b(|x_{i+2}| + \cdots + |x_n|), \\
|g_i(\widetilde{x}_{i+2})| &\le b(|x_{i+2}| + \cdots + |x_n|).
\end{aligned}
\tag{3}
$$

**Assumption 2.** *Suppose that the symbol for $d_i(t), t \in R_+$ is positive, and for $i = 1, \cdots, n$, there are unknown normal numbers $\lambda_i$, $\mu_i$, and $\mu$, such that*

$$
0 < \lambda_i \le d_i(t) \le \mu_i \le \mu.
\tag{4}
$$

**Remark 1.** *According to Assumption 1, the nonlinear terms $f_i(\widetilde{x}_{i+2})$ and $g_i(\widetilde{x}_{i+2})$ of system (2) depend on $x_{i+2}, \cdots, x_n$, which results the upper-triangular structure. As discussed in deterministic upper-triangular systems [35,41,42] and stochastic upper-triangular systems [36,37,43,44], Assumption 1 is a reasonable assumption. Assumption 2 shows that $d_i(t)$ is time-varying, and that both upper and lower bounds are unknown, which is more general than systems [36–39] with deterministic control coefficients.*

### 3.1. State Feedback Control of Nominal Systems

We first consider the state feedback control of nominal systems:

$$
\begin{aligned}
dz_i &= d_i(t)z_{i+1}dt, \ i = 1, \cdots, n-1, \\
dz_n &= d_n(t)vdt.
\end{aligned}
\tag{5}
$$

Next, we design an appropriate state feedback controller using the backstepping method and then conduct a stability analysis.

**Step 1.** Choosing a coordinate change $\xi_1 = z_1$ and the Lyapunov function $V_1(\bar{z}_1) = \frac{1}{4}\xi_1^4$ from (5), it follows that

$$
\mathcal{L}V_1(\bar{z}_1) \le \frac{\partial V_1}{\partial z_1}d_1 z_2 \le d_1 z_1^3(z_2 - z_2^*) + d_1 z_1^3 z_2^*.
\tag{6}
$$

With Assumption 2, we have

$$d_1 z_1^3 z_2^* \leq \mu z_1^3 z_2^*. \tag{7}$$

Then, by adding the term $\mu c_1 \xi_1^4$ to the right side of Equation (6) and then subtracting it, and by utilizing (7), we get

$$\mathcal{L}V_1(\bar{z}_1) \leq -\mu c_1 \xi_1^4 + \mu c_1 \xi_1^4 + d_1 \xi_1^3 (z_2 - z_2^*) + \mu \xi_1^3 z_2^*, \tag{8}$$

where $c_1 > 0$ is a design parameter.

Clearly, selecting the virtual controller

$$z_2^* = -c_1 \xi_1 := -\alpha_1 \xi_1 \tag{9}$$

results in

$$\mathcal{L}V_1(\bar{z}_1) \leq -\mu c_1 \xi_1^4 + d_1 \xi_1^3 (z_2 - z_2^*). \tag{10}$$

**Step 2.** In choosing $\xi_2 = z_2 - z_2^*$ and the Lyapunov function $V_2(\bar{z}_2) = V_1(\bar{z}_1) + \frac{1}{4}\xi_2^4$, we can easily derive

$$\mathcal{L}V_2(\bar{z}_2) \leq -\mu c_1 \xi_1^4 + d_1 \xi_1^3 \xi_2 + d_2 \xi_2^3 (z_3 - z_3^*) + d_2 \xi_2^3 z_3^* - d_1 \xi_2^3 \frac{\partial z_2^*}{\partial z_1} z_2. \tag{11}$$

By Lemma A4 and Assumption 2, we have

$$d_1 \xi_1^3 \xi_2 \leq \mu \frac{3}{4} \varepsilon_{211} \xi_1^4 + \mu \frac{1}{4} \varepsilon_{211}^{-3} \xi_2^4, \tag{12}$$

and

$$
\begin{aligned}
-d_1 \xi_2^3 \frac{\partial z_2^*}{\partial z_1} z_2 &= -d_1 \xi_2^3 \frac{\partial z_2^*}{\partial z_1} (\xi_2 - \alpha_1 \xi_1) \\
&\leq \mu \left| \frac{\partial z_2^*}{\partial z_1} \right| \xi_2^4 + \mu \left| \frac{\partial z_2^*}{\partial z_1} \right| |\xi_2|^3 |\xi_1| \alpha_1 \\
&\leq \mu \frac{1}{4} \bar{\varepsilon}_{212} \xi_1^4 + \mu \sqrt{1 + \left( \frac{\partial z_2^*}{\partial z_1} \right)^2} \xi_2^4 + \mu \frac{3}{4} \bar{\varepsilon}_{212}^{-\frac{1}{3}} \left( \alpha_1 \sqrt{1 + \left( \frac{\partial z_2^*}{\partial z_1} \right)^2} \right)^{\frac{4}{3}} \xi_2^4, \quad (13)
\end{aligned}
$$

where $\varepsilon_{211} > 0$ and $\bar{\varepsilon}_{212} > 0$ are design parameters.

According to (12) and (13), normal numbers $c_{21}$ and $H_{21}$ are defined as

$$c_{21} = \frac{3}{4} \varepsilon_{211} + \frac{1}{4} \bar{\varepsilon}_{212},$$

$$H_{21} = \frac{1}{4} \varepsilon_{211}^{-3} + \sqrt{1 + \left( \frac{\partial z_2^*}{\partial z_1} \right)^2} + \frac{3}{4} \bar{\varepsilon}_{212}^{-\frac{1}{3}} \left( \alpha_1 \sqrt{1 + \left( \frac{\partial z_2^*}{\partial z_1} \right)^2} \right)^{\frac{4}{3}}. \tag{14}$$

Based on Assumption 2, we have

$$d_2 z_2^3 z_3^* \leq \mu z_2^3 z_3^*. \tag{15}$$

Therefore, substituting (12)–(15) into (11) results in

$$\mathcal{L}V_2(\bar{z}_2) \leq -\mu c_1 \xi_1^4 + d_2 \xi_2^3 (z_3 - z_3^*) + d_2 \xi_2^3 z_3^* + \mu c_{21} \xi_1^4 + \mu H_{21} \xi_2^4. \tag{16}$$

Adding the term $\mu c_2 \xi_2^4$ to the right side of Equation (16) and then subtracting it results in

$$\mathcal{L}V_2(\bar{z}_2) \leq d_2\xi_2^3(z_3 - z_3^*) + \mu\xi_2^3 z_3^* - \mu(c_1 - c_{21})\xi_1^4 + \mu(c_2 + H_{21})\xi_2^4 - \mu c_2\xi_2^4, \quad (17)$$

where $c_2 > 0$ is the selected design parameter, and $c_1 - c_{21} > 0$ by choosing the appropriate design parameters.

The smooth virtual control $z_3^*$ is shown below:

$$z_3^* = -(c_2 + H_{21})\xi_2 := -\alpha_2\xi_2, \quad (18)$$

and by substituting (18) into (17), we have

$$\mathcal{L}V_2(\bar{z}_2) \leq d_2\xi_2^3(z_3 - z_3^*) - \mu(c_1 - c_{21})\xi_1^4 - \mu c_2\xi_2^4. \quad (19)$$

**Step i.** Suppose that at step $i - 1$, there exists a positive definite and $\mathcal{C}^2$ Lyapunov function $V_{i-1}(\bar{z}_{i-1})$ as well as a set of definitions for the following virtual controllers $z_2^*, \cdots, z_i^*$

$$z_2^* = -\alpha_1\xi_1,$$

$$\vdots$$

$$z_i^* = -\alpha_{i-1}\xi_{i-1}, \quad (20)$$

where $\xi_j = z_j - z_j^*$ $(1 \leq j \leq i - 1)$ and $\alpha_k$ $(1 \leq k \leq i - 1)$ are normal numbers; thus,

$$\mathcal{L}V_{i-1}(\bar{z}_{i-1}) \leq -\mu \sum_{j=1}^{i-1}\left(c_j - \sum_{k=j+1}^{i-1} c_{kj}\right)\xi_j^4 + d_{i-1}\xi_{i-1}^3(z_i - z_i^*), \quad (21)$$

where $c_j - \sum_{k=j+1}^{i-1} c_{kj} > 0$ by choosing the appropriate design parameters.

To continue with the induction, we can select $\xi_i = z_i - z_i^*$ and the following Lyapunov function in the $i$th step:

$$V_i(\bar{z}_i) = V_{i-1}(\bar{z}_{i-1}) + \frac{1}{4}\xi_i^4, \quad (22)$$

where $\bar{z}_i = (z_1, \cdots, z_i)^T$. From (21) and (22), we get

$$\mathcal{L}V_i(\bar{z}_i) \leq -\mu \sum_{j=1}^{i-1}\left(c_j - \sum_{k=j+1}^{i-1} c_{kj}\right)\xi_j^4 + d_{i-1}\xi_{i-1}^3\xi_i$$

$$+ d_i\xi_i^3(z_{i+1} - z_{i+1}^*) + d_i\xi_i^3 z_{i+1}^* - \xi_i^3 \sum_{k=1}^{i-1} \frac{\partial z_{i+1}}{\partial z_k} d_k z_{k+1}. \quad (23)$$

From (20), Assumption 2, and Lemma A4, we can conclude that

$$
d_{i-1}\xi_{i-1}^3\xi_i \leq \mu\frac{3}{4}\varepsilon_{i,i-1,1}\xi_{i-1}^4 + \mu\frac{1}{4}\varepsilon_{i,i-1,1}^{-3}\xi_i^4,
$$

$$
\begin{aligned}
-\xi_i^3\sum_{k=1}^{i-1}\frac{\partial z_i^*}{\partial z_k}d_k z_{k+1} &\leq \mu|\xi_i|^3\sum_{k=1}^{i-2}\left|\frac{\partial z_i^*}{\partial z_k}\right||\xi_{k+1}| + \mu|\xi_i|^3\sum_{k=1}^{i-2}\left|\frac{\partial z_i^*}{\partial z_k}\right||\alpha_k\xi_k| \\
&\quad + \mu\left|\frac{\partial z_i^*}{\partial z_{i-1}}\right||\xi_i|^4 + \mu|\xi_i|^3\left|\frac{\partial z_i^*}{\partial z_{i-1}}\right||\alpha_{i-1}\xi_{i-1}| \\
&\leq \mu\sum_{k=2}^{i-2}\frac{1}{4}\varepsilon_{ik2}\xi_k^4 + \mu\sum_{k=1}^{i-2}\frac{3}{4}\varepsilon_{i,k+1,2}^{-\frac{1}{3}}\left(\sqrt{1+\left(\frac{\partial z_i^*}{\partial z_k}\right)^2}\right)^{\frac{4}{3}}\xi_i^4 \\
&\quad + \mu\sum_{k=1}^{i-2}\frac{1}{4}\bar{\varepsilon}_{ik2}\xi_k^4 + \mu\sum_{k=1}^{i-2}\frac{3}{4}\bar{\varepsilon}_{ik2}^{-\frac{1}{3}}\left(\alpha_k\sqrt{1+\left(\frac{\partial z_i^*}{\partial z_k}\right)^2}\right)^{\frac{4}{3}}\xi_i^4 \\
&\quad + \mu\sqrt{1+\left(\frac{\partial z_i^*}{\partial z_{i-1}}\right)^2}\xi_i^4 + \mu\frac{1}{4}\bar{\varepsilon}_{i,i-1,2}\xi_{i-1}^4 \\
&\quad + \mu\frac{3}{4}\bar{\varepsilon}_{i,i-1,2}^{-\frac{1}{3}}\left(\alpha_{i-1}\sqrt{1+\left(\frac{\partial z_i^*}{\partial z_{i-1}}\right)^2}\right)^{\frac{4}{3}}\xi_i^4.
\end{aligned}
\tag{24}
$$

By (24), the normal numbers $c_{ik}$ and $H_{i1}$ can be defined as follows:

$$
c_{i1} = \frac{1}{4}\bar{\varepsilon}_{i12},
$$

$$
c_{ik} = \frac{1}{4}\bar{\varepsilon}_{ik2} + \frac{1}{4}\varepsilon_{ik2}, k = 2,\cdots,i-2,
$$

$$
c_{i,i-1} = \frac{3}{4}\varepsilon_{i,i-1,1} + \frac{1}{4}\varepsilon_{i,i-1,2} + \frac{1}{4}\bar{\varepsilon}_{i,i-1,2},
$$

$$
\begin{aligned}
H_{i1} &= \frac{1}{4}\varepsilon_{i,i-1,1}^{-3} + \sum_{k=1}^{i-2}\frac{3}{4}\varepsilon_{i,k+1,2}^{-\frac{1}{3}}\left(\sqrt{1+\left(\frac{\partial z_i^*}{\partial z_k}\right)^2}\right)^{\frac{4}{3}} \\
&\quad + \sum_{k=1}^{i-2}\frac{3}{4}\bar{\varepsilon}_{ik2}^{-\frac{1}{3}}\left(\alpha_k\sqrt{1+\left(\frac{\partial z_i^*}{\partial z_k}\right)^2}\right)^{\frac{4}{3}} + \sqrt{1+\left(\frac{\partial z_i^*}{\partial z_{i-1}}\right)^2} \\
&\quad + \frac{3}{4}\bar{\varepsilon}_{i,i-1,2}^{-\frac{1}{3}}\left(\alpha_{i-1}\sqrt{1+\left(\frac{\partial z_i^*}{\partial z_{i-1}}\right)^2}\right)^{\frac{4}{3}},
\end{aligned}
\tag{25}
$$

where $\varepsilon_{i,i-1,j}$ $(j=1,2)$ and $\varepsilon_{ik2}$ $(k=1,\cdots,i-1)$ are normal numbers.

By adding and subtracting $\mu c_i\xi_i^4$ on the right side of (23) and then using (24) and (25) in (23) results in

$$
\begin{aligned}
\mathcal{L}V_i(\bar{z}_i) &\leq -\mu\sum_{j=1}^{i-1}\left(c_j - \sum_{k=j+1}^{i-1}c_{kj}\right)\xi_j^4 + d_i\xi_i^3(z_{i+1}-z_{i+1}^*) + d_i\xi_i^3 z_{i+1}^* \\
&\quad + \mu\sum_{k=1}^{i-1}c_{ik}\xi_k^4 + \mu H_{i1}\xi_i^4 + \mu c_i\xi_i^4 - \mu c_i\xi_i^4.
\end{aligned}
\tag{26}
$$

Clearly, by choosing the virtual controller as

$$
z_{i+1}^* = -(c_i + H_{i1})\xi_i := -\alpha_i\xi_i,
\tag{27}
$$

we get

$$\mathcal{L}V_i(\bar{z}_i) \le -\mu \sum_{j=1}^{i}\left(c_j - \sum_{k=j+1}^{i} c_{kj}\right)\xi_j^4 + d_i\xi_i^3(z_{i+1} - z_i^*), \tag{28}$$

where $c_j - \sum_{k=j+1}^{i} c_{kj} > 0$ by choosing the appropriate design parameters.

**Step n.** Consider the Lyapunov function for system (5):

$$V_n(\bar{z}_n) = V_{n-1}(\bar{z}_{n-1}) + \frac{1}{4}\xi_n^4, \tag{29}$$

where $\xi_n = z_n - z_n^*$. By (23), we have

$$\begin{aligned}
\mathcal{L}V_n(\bar{z}_n) \le &-\mu \sum_{j=1}^{n-1}\left(c_j - \sum_{k=j+1}^{n-1} c_{kj}\right)\xi_j^4 + d_n\xi_n^3 v \\
&+ \mu \sum_{k=1}^{n-1} c_{nk}\xi_k^4 + \mu H_{n1}\xi_n^4 + \mu c_n\xi_n^4 - \mu c_n\xi_n^4,
\end{aligned} \tag{30}$$

where $c_j - \sum_{k=j+1}^{n-1} c_{kj} > 0$ by choosing the appropriate design parameters. Obviously, designing the state feedback control law

$$v = -(c_n + H_{n1})\xi_n := -\alpha_n\xi_n \tag{31}$$

yields

$$\mathcal{L}V_n(\bar{z}_n) \le -\mu \sum_{j=1}^{n}\left(c_j - \sum_{k=j+1}^{n} c_{kj}\right)\xi_j^4, \tag{32}$$

where $c_j - \sum_{k=j+1}^{n} c_{kj} > 0$ by choosing the appropriate design parameters.

### 3.2. State Feedback Control and Stability Analysis

Using the results in Section 3.1, the following main results can be obtained.

**Theorem 1.** *Under the condition of Assumption 1, the upper-triangular stochastic nonlinear system (2) with time-varying control coefficients can achieve global asymptotical stability in probability by the state feedback controller.*

**Proof of Theorem 1.** First of all, a coordinate transformation is introduced:

$$z_i = \frac{x_i}{\varepsilon^{i-1}}, v = \frac{u}{\varepsilon^n}, i = 1, \cdots, n, \tag{33}$$

where $0 < \varepsilon < 1$ is an undetermined parameter. Then, we have

$$\begin{aligned}
dz_1 &= \left(\varepsilon d_1(t)z_2 + \bar{f}_1(\tilde{z}_3)\right)dt + \bar{g}_1^T(\tilde{z}_3)d\omega, \\
dz_2 &= \left(\varepsilon d_2(t)z_3 + \bar{f}_2(\tilde{z}_4)\right)dt + \bar{g}_2^T(\tilde{z}_4)d\omega, \\
&\vdots \\
dz_{n-2} &= \left(\varepsilon d_{n-2}(t)z_{n-1} + \bar{f}_{n-2}(\tilde{z}_n)\right)dt + \bar{g}_{n-2}^T(\tilde{z}_n)d\omega, \\
dz_{n-1} &= \varepsilon d_{n-1}(t)z_n dt, \\
dz_n &= \varepsilon d_n(t)v dt,
\end{aligned} \tag{34}$$

where $\bar{f}_i(\widetilde{z}_{i+2}) = \frac{f_i(\widetilde{x}_{i+2})}{\varepsilon^{i-1}}, \bar{g}_i(\widetilde{z}_{i+2}) = \frac{g_i(\widetilde{x}_{i+2})}{\varepsilon^{i-1}}$ and $\widetilde{z}_i = (z_i, \cdots, z_n)^T$. System (34) can then be written in compact form:

$$dz = \varepsilon E dt + F dt + G^T d\omega, \tag{35}$$

where

$$
\begin{aligned}
z &= (z_1, \cdots, z_n)^T, \\
E(z) &= (d_1 z_2, \cdots, d_{n-1} z_n, d_n v)^T, \\
F(z) &= (\bar{f}_1(\widetilde{z}_3), \cdots, \bar{f}_{n-2}(\widetilde{z}_n), 0, 0)^T, \\
G^T(z) &= (\bar{g}_1(\widetilde{z}_3), \cdots, \bar{g}_{n-2}(\widetilde{z}_n), 0, 0)^T.
\end{aligned} \tag{36}
$$

In this paper, $E_i(z)$ and $F_i(z)$ respectively represent the $i$th element of $E(z)$ and $F(z)$, and $G_i(z)$ represents the $i$th element of $G_i(z)$. Thus, by (5), (29), and (32), $\frac{\partial V_n}{\partial z} E(z)$ is negative definite. Given a dilation weight of $\Delta = (1, 1, \cdots, 1)$ and $\frac{\partial V_n}{\partial z} E(z) = \sum_{i=1}^{n} \frac{\partial V_n}{\partial z_i} E_i(z)$, $\frac{\partial V_n}{\partial z_i}$ and $E_i(z)$ are homogeneous of degree 3 and 1, respectively. By Lemmas A2 and A3, we can obtain

$$\frac{\partial V_n}{\partial z} E(z) \leq -\mu \underline{c}_0 \|z\|_{\triangle}^4, \tag{37}$$

for a constant $\underline{c}_0 > 0$ and $\|z\|_{\triangle}^4 = (\sum_{i=1}^{n} |z_i|^2)^2$. In view of $0 < \varepsilon < 1$, by using Assumption 1 and (33), we get

$$
\begin{aligned}
|\bar{f}_i(\widetilde{z}_{i+2})| &= \left| \frac{f_i(\widetilde{x}_{i+2})}{\varepsilon^{i-1}} \right| \leq b\varepsilon^2 \sum_{j=i+2}^{n} |z_j|, \\
|\bar{g}_i(\widetilde{z}_{i+2})| &= \left| \frac{g_i(\widetilde{x}_{i+2})}{\varepsilon^{i-1}} \right| \leq b\varepsilon^2 \sum_{j=i+2}^{n} |z_j|.
\end{aligned} \tag{38}
$$

Similarly, by (38) and Lemmas A2 and A3, we have

$$\frac{\partial V_n}{\partial z} F(z) = \sum_{i=1}^{n} \frac{\partial V_n}{\partial z_i} F_i(z) \leq -c_0 \varepsilon^2 \|z\|_{\triangle}^4, \tag{39}$$

where $c_0 > 0$ is a number.

According Lemma A3, we know that $\frac{\partial^2 V_n}{\partial z_i \partial z_j}$ is homogeneous of degree 2. Noting that $G(z) \in R^{r \times n}$, and by using (37) and Lemmas A2 and A3, we can obtain

$$
\begin{aligned}
\frac{1}{2} \text{Tr} \left\{ G(z) \frac{\partial^2 V_n}{\partial z^2} G^T(z) \right\} &\leq \frac{1}{2} r \left| G(z) \frac{\partial^2 V_n}{\partial z^2} G^T(z) \right|_{\infty} \\
&\leq \frac{1}{2} r \sqrt{r} \left| G(z) \frac{\partial^2 V_n}{\partial z^2} G^T(z) \right| \\
&\leq \frac{1}{2} r \sqrt{r} \left| \sum_{i,j=1}^{n} G_i(z) \frac{\partial^2 V_n}{\partial z_i \partial z_j} G_j^T(z) \right| \\
&\leq \frac{1}{2} r \sqrt{r} \sum_{i,j=1}^{n} \left| G_i(z) \frac{\partial^2 V_n}{\partial z_i \partial z_j} G_j^T(z) \right| \\
&\leq \frac{1}{2} r \sqrt{r} \sum_{i,j=1}^{n} \left| \frac{\partial^2 V_n}{\partial z_i \partial z_j} \right| \cdot |G_i(z)| \cdot \left| G_j^T(z) \right| \\
&\leq \bar{c}_0 \varepsilon^2 \|z\|_{\triangle}^4,
\end{aligned} \tag{40}
$$

where $\bar{c}_0 > 0$ is a constant, and the second equation is obtained by utilizing $|A|_\infty \leq \sqrt{r}|A|$ ($A$ is a square matrix with an $r$-dimension).

For system (35), with help from (37)–(40), the following is derived:

$$
\begin{aligned}
\mathcal{L}V_n(\bar{z}_n)|_{(35)} &= \varepsilon \frac{\partial V_n}{\partial z}E(z) + \frac{\partial V_n}{\partial z}F(z) + \frac{1}{2}\text{Tr}\left\{G(z)\frac{\partial^2 V_n}{\partial z^2}G^T(z)\right\} \\
&\leq -\mu\varepsilon\underline{c}_0\|z\|_\triangle^4 + c_0\varepsilon^2\|z\|_\triangle^4 + \bar{c}_0\varepsilon^2\|z\|_\triangle^4 \\
&\leq -\varepsilon(\mu\underline{c}_0 - \varepsilon(c_0 + \bar{c}_0)\|z\|_\triangle^4,
\end{aligned}
\tag{41}
$$

Obviously, under the condition that the gain $\varepsilon$ is sufficiently small, the right side of (41) is negative definite. Therefore, $\varepsilon$ is sufficiently small to make the following formula valid:

$$
\mathcal{L}V_n(\bar{z}_n)|_{(35)} \leq -c\|z\|_\triangle^4,
\tag{42}
$$

where $c > 0$ is a constant. According Lemma A1, system (35) is GAS in probability. It can be known from (33) that system (2) is GAS in probability. □

**Remark 2.** *The reason why we firstly use the backstepping technique to tackle the stability problem of the nominal systems is that it is so hard to study the stability of the original nonlinear systems directly. Consequently, a low-gain homogeneous dominant control strategy is proposed for the state feedback stability of upper-triangular stochastic nonlinear systems with time-varying control coefficients. In this process, we scale the uncertain control coefficients $d_i(t)$ to supper bound and use Young's inequality many times to tackle the stability problem of the system.*

**Remark 3.** *Since the structure of the drift and diffusion terms is symmetric, we use a low-gain in the controller to remove the influence of the drift and diffusion terms. Compared with the existing upper-triangular stochastic nonlinear systems, we extend the low-gain homogeneous dominance technique from deterministic systems to time-varying systems.*

**Remark 4.** *The strict proof of Theorem 1 is not a simple task, but it also involves the verification of Lemma A1 conditions.*

## 4. A Simulation Example

In this section, a numerical example is used to verify the rationality and validity of the results in Section 3.

Suppose that the following system is taken for simulation:

$$
\begin{aligned}
dx_1 &= \left((5 + 0.1\sin t)x_2 + x_3\cos^2 x_3\right)dt + x_3\cos x_3 d\omega, \\
dx_2 &= (3 + 2\sin t)x_3 dt, \\
dx_3 &= (2 + 0.5\sin t)u dt.
\end{aligned}
\tag{43}
$$

Obviously, the system satisfies Assumption 1. Notice that there exist normal numbers $\lambda_i$, $\mu_i(i = 1, 2, 3)$, and $\mu$, with $\lambda_1 \leq 4.9 \leq d_1(t) \leq 5.1 \leq \mu_1 \leq \mu$, $\lambda_2 \leq 1 \leq d_2(t) \leq 5 \leq \mu_2 \leq \mu$ and $\lambda_3 \leq 1.5 \leq d_3(t) \leq 2.5 \leq \mu_3 \leq \mu$, which satisfy Assumption 2.

Inputting the coordinates

$$
z_1 = x_1, z_2 = \frac{x_2}{\varepsilon}, z_3 = \frac{x_3}{\varepsilon^2}, v = \frac{u}{\varepsilon^3},
\tag{44}
$$

where $0 < \varepsilon < 1$ is an undetermined parameter, system (43) could be rewritten as

$$
\begin{aligned}
dz_1 &= \left(\varepsilon(5 + 0.1\sin t)z_2 + \varepsilon^2 z_3\cos^2 \varepsilon^2 z_3\right)dt + \varepsilon^2 z_3\cos \varepsilon^2 z_3 d\omega, \\
dz_2 &= \varepsilon(3 + 2\sin t)z_3 dt, \\
dz_3 &= \varepsilon(2 + 0.5\sin t)v dt.
\end{aligned}
\tag{45}
$$

According to the design process above, the controller can be obtained as follows:

$$v = -598(60z_1 + 4z_2 + z_3).$$
(46)

By (44), we can get the controller as follows:

$$u = -598(60\varepsilon^3 x_1 + 4\varepsilon^2 x_2 + \varepsilon x_3).$$
(47)

The simulation is performed by selecting $\varepsilon = 0.1$, $x_1(0) = 1$, $x_2(0) = 0.6$, and $x_3(0) = 0.3$. Figure 1 illustrates the responses of the closed-loop systems (43)–(47) and verifies the effectiveness of the controller.

**Remark 5.** *The simulation shows only a numerical example, not a real example. As can be seen in Figure 1, under the designed controller, the response curves of the closed-loop system almost certainly converge to zero. In general, when there is a time-varying coefficient, the controller designed based on the backstepping technique and the low-gain homogeneous domination technology has a better performance, so it is of great significance in practical applications. Now that we know the importance of physical models, finding a real-world mechanical device that can be modeled directly with system (2) or that can be transformed into a system (2) through coordinate transformation is a top priority for future research.*

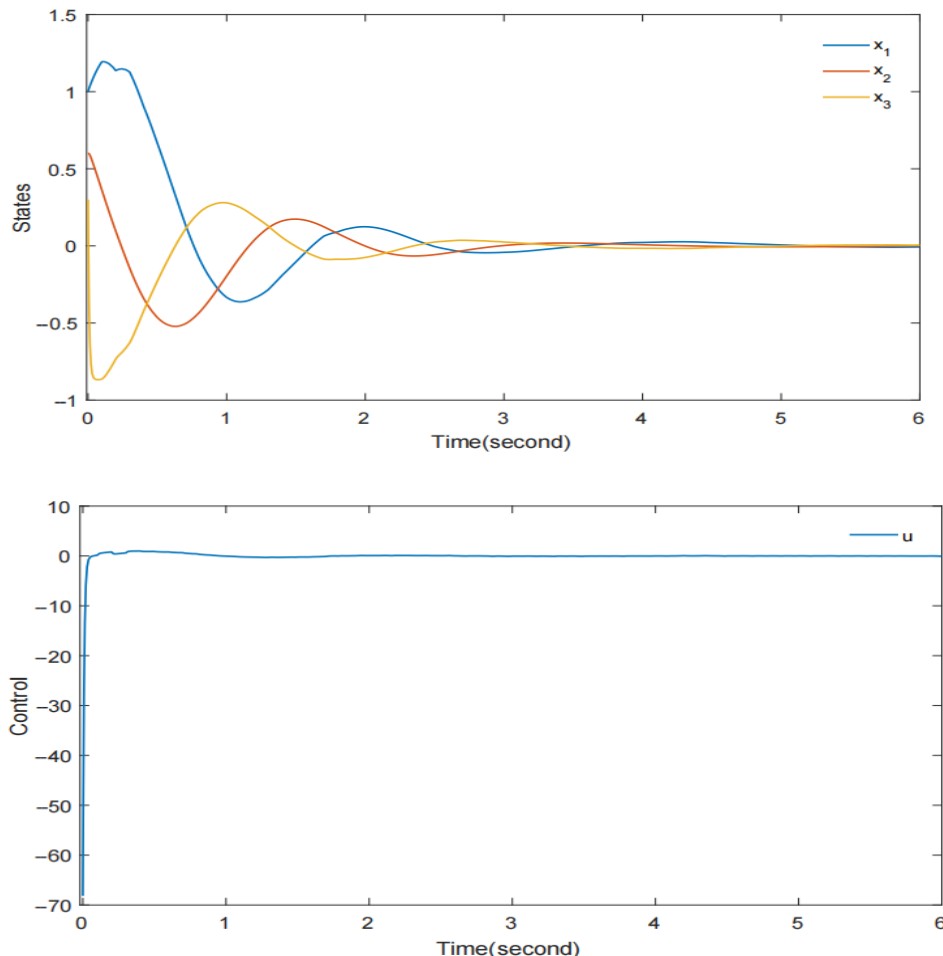

**Figure 1.** The responses of a closed-loop system.

**Remark 6.** *It should be further noted that the difference between this paper and other papers is that this is a theoretical study that provides a new idea for solving the state feedback stabilization problem of trigonometric stochastic nonlinear systems with time-varying control coefficients.*

## 5. Conclusions

The state feedback control problem is studied in this article by utilizing the backstepping technique, the low-gain homogeneous domination technique, and some significant inequalities. The systems are considered to be upper-triangular stochastic nonlinearities, and the control coefficients are uncertain. The designed controller is able to ensure that the closed-loop system is GAS almost everywhere. Based on these, there are some associated issues that could be researched more in the future. For instance, the extension of this control strategy to more general systems, such as high-order stochastic systems, should be considered. A practical example for system (2) for simulation verification should also be found. The results of fixed points in b-Metric Space can then be considered for stochastic nonlinear systems [45,46].

**Author Contributions:** Conceptualization, X.S. and H.Y.; methodology and software, X.S. and X.Y.X.; writing and editing, X.S., H.Y. and X.X.; funding acquisition, H.Y. All authors have read and agreed to the published version of the manuscript.

**Funding:** This research was funded by Shandong Province Social Science Planning Research Project of China, Project number (21CSDJ21); Shandong Provincial Graduate Education and Teaching Reform Research Project of China, Project number (SDYJG19073).

**Institutional Review Board Statement:** Not applicable.

**Informed Consent Statement:** Not applicable.

**Data Availability Statement:** Not applicable.

**Conflicts of Interest:** The authors declare no conflict of interest.

## Appendix A

**Lemma A1** ([2])**.** *Considering stochastic system (1), if there exist a $C^2$ function $V(x)$, class $\mathcal{K}_\infty$ functions $\beta_1$ and $\beta_2$, a constant $c > 0$, and a nonnegative function $W(x)$, such that*

$$\beta_1(|x|) \leq V(x) \leq \beta_2(|x|), \quad \mathcal{L}V \leq -cW(x),$$

*then the following conclusions hold:*

*(1) For (1), there exists an almost surely unique solution on $[0, \infty)$ for any $x_0$;*

*(2) When $f(0) = 0$, $g(0) = 0$, and $W(x) = \beta_3(|x|)$ is continuous, the equilibrium $x = 0$ is GAS in probability and $P\{\lim_{t\to\infty} W(x(t)) = 0\} = 1$, where $\beta_3(\cdot)$ is a class $\mathcal{K}$ function.*

**Lemma A2** ([35])**.** *Given a dilation weight $\triangle = (r_1, \cdots, r_n)$, suppose that $V_1(x)$ and $V_2(x)$ are homogeneous functions of degrees $\tau_1$ and $\tau_2$, respectively. Then, $V_1(x)V_2(x)$ is also homogeneous with respect to the same dilation weight $\triangle$. Moreover, the homogeneous degree of $V_1 \cdot V_2$ is $\tau_1 + \tau_2$.*

**Lemma A3** ([35])**.** *Suppose that $V : R^n \to R$ is a homogeneous function of degree $\tau$ with respect to the dilation weight $\triangle$. Then, the following hold:*

*(1) $\frac{\partial V}{\partial x_i}$ is homogeneous of degree $\tau - r_i$, with $r_i$ being the homogeneous weight of $x_i$;*

*(2) There is a constant $\bar{c}$ such that*

$$V(x) \leq \bar{c}\|x\|_{\triangle}^{\tau}.$$

*Moreover, if $V(x)$ is positive definite, then*

$$\underline{c}\|x\|_{\triangle}^{\tau} \leq V(x),$$

*where $\underline{c} > 0$ is a constant.*

**Lemma A4** ([47]). *Let* $x, y$ *be real variables, for any positive integers m, n, and any real number* $\varepsilon > 0$, *the following inequality holds:*

$$|x|^m |y|^n \leq \frac{m}{m+n} \varepsilon |x|^{m+n} + \frac{n}{m+n} \varepsilon^{-\frac{m}{n}} |y|^{m+n}.$$

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
