# Peer review of "State Feedback Stabilization for a Class of Upper-Triangular Stochastic Nonlinear Systems with Time-Varying Control Coefficients"

_processes, doi:10.3390/pr10081465_

Round 1
Reviewer 2 Report
1- Most of Math calculations and even Lemma should be packed in an appendix to make the paper more scientifically interesting to readers other than pure Math paper.
2- The way authors validate their designed controller using an example looks vague. It's better to apply their controller to a real system to show how accurate their controller works.
3- In line 134 and 135, authors should represent their meaning by claiming general systems. What sections in a general system is missing here which would be generalized in future work?
Reviewer 3 Report
Review: processes-1800796 Title: State feedback control of a class of upper-triangular stochastic nonlinear systems with uncertain control coefficientsThe paper deals with the control of upper-triangular stochastic nonlinear systems with uncertain control coefficients via state feedback with backstepping.
The description of the controllers appears to be well designed, and the results appear to be coherent. I generally feel the paper is interesting with solid results that can be extended to many applications.
Generally, the manuscript is well-written, and the complete literature review is highly appreciated.
Author Response
We supplement the introduction with stochastic nonlinear systems of upper-triangle and lower-triangle.
Thanks for professor's appreciation, we will continue to work hard to create better papers. Everything goes well. Your kind considerations will be greatly appreciated.
Round 2
Reviewer 1 Report
The revised version of this paper is very good. The author proved all suggested comments about the paper. So, I strongly recommend it for publication.